# Metabolic Profiling, Chemical Composition, Antioxidant Capacity, and In Vivo Hepato- and Nephroprotective Effects of *Sonchus cornutus* in Mice Exposed to Cisplatin

**DOI:** 10.3390/antiox11050819

**Published:** 2022-04-22

**Authors:** Sameh S. Elhady, Reda F. A. Abdelhameed, Eman T. Mehanna, Alaa Samir Wahba, Mahmoud A. Elfaky, Abdulrahman E. Koshak, Ahmad O. Noor, Hanin A. Bogari, Rania T. Malatani, Marwa S. Goda

**Affiliations:** 1Department of Natural Products, Faculty of Pharmacy, King Abdulaziz University, Jeddah 21589, Saudi Arabia; melfaky@kau.edu.sa (M.A.E.); aekoshak@kau.edu.sa (A.E.K.); 2Department of Pharmacognosy, Faculty of Pharmacy, Galala University, New Galala 43713, Egypt; 3Department of Pharmacognosy, Faculty of Pharmacy, Suez Canal University, Ismailia 41522, Egypt; marwa_saeed@pharm.suez.edu.eg; 4Department of Biochemistry, Faculty of Pharmacy, Suez Canal University, Ismailia 41522, Egypt; eman.taha@pharm.suez.edu.eg (E.T.M.); alaa.samir@pharm.suez.edu.eg (A.S.W.); 5Centre for Artificial Intelligence in Precision Medicines, King Abdulaziz University, Jeddah 21589, Saudi Arabia; 6Department of Pharmacy Practice, Faculty of Pharmacy, King Abdulaziz University, Jeddah 21589, Saudi Arabia; aonoor@kau.edu.sa (A.O.N.); hbogari@kau.edu.sa (H.A.B.); rmalatani@kau.edu.sa (R.T.M.)

**Keywords:** *Sonchus cornutus*, polyphenolic compounds, hepatotoxicity, nephrotoxicity, oxidative stress, inflammation, apoptosis, drug discovery, industries development

## Abstract

*Sonchus cornutus* (Asteraceae) is a wild. edible plant that represents a plentiful source of polyphenolic compounds. For the first time, the metabolic analysis profiling demonstrated the presence of anthocyanidin glycosides, coumarins, flavonoids and their corresponding glycosides, and phenolic acids. The total phenolic compounds were determined to be 206.28 ± 14.64 mg gallic acid equivalent/gm, while flavonoids were determined to be 45.56 ± 1.78 mg quercetin equivalent/gm. The crude extract of *S. cornutus* exhibited a significant 1,1-diphenyl-2-picrylhydrazyl free radical scavenging effect with half-maximal inhibitory concentration (IC_50_) of 16.10 ± 2.14 µg/mL compared to ascorbic acid as a standard (10.64 ± 0.82 µg/mL). In vitro total antioxidant capacity and ferric reducing power capacity assays revealed a promising reducing potential of *S. cornutus* extract. Therefore, the possible protective effects of *S. cornutus* against hepatic and renal toxicity induced by cisplatin in experimental mice were investigated. *S. cornutus* significantly ameliorated the cisplatin-induced disturbances in liver and kidney functions and oxidative stress, decreased MDA, ROS, and NO levels, and restored CAT and SOD activities. Besides, it reversed cisplatin-driven upregulation in inflammatory markers, including iNOS, IL-6, and IL-1β levels and NF-κB and TNF-α expression, and elevated anti-inflammatory IL-10 levels and Nrf2 expression. Additionally, the extract mitigated cisplatin alteration in apoptotic (Bax and caspase-3) and anti-apoptotic (Bcl-2) proteins. Interestingly, hepatic, and renal histopathology revealed the protective impacts of *S. cornutus* against *cisplatin*-induced pathological changes. Our findings guarantee a protective effect of *S. cornutus* against cisplatin-induced hepatic and renal damage via modulating oxidative stress, inflammation, and apoptotic pathways.

## 1. Introduction

Cisplatin, a platinum-containing drug, is one of the most widely used chemotherapeutic agents in the management of various types of cancer. The anticancer effect of cisplatin is thought to be attributable to DNA adducts formation and G2 cell cycle arrest, eventually leading to apoptosis. However, its application and effectiveness are limited due to drug resistance as a result of reduced drug accumulation in cancer cells, glutathione, and metallothioneins-induced drug inactivation and accelerated repair of DNA lesions [1]. Another inherent challenge of cisplatin is that the dose scale required to overwhelm even a minor increase in cellular resistance can adversely affect normal cells and cause severe cytotoxicity, with nephrotoxicity being considered the principal dose-limiting side effect. More to the point, massive hepatic toxicity can occur with repeated higher doses of cisplatin [2].

Cisplatin-induced cytotoxicity has been attributed to enhanced oxidative stress and increased production of reactive oxygen species (ROS) which deplete glutathione levels and reduce activity of the antioxidant enzymes superoxide dismutase (SOD) and catalase. Moreover, ROS can impair cell integrity via lipid peroxidation, protein denaturation, and DNA damage [3]. In addition, involvement of proinflammatory genes such as tumor necrosis factor-α (TNF-α), inducible nitric oxide synthase (iNOS), mitochondrial dysfunction, disturbed Ca^2+^ homeostasis, and apoptosis have been demonstrated to exert crucial role in the mechanisms associated with cisplatin-induced cytotoxicity [4,5]. Considering that oxidative stress is a key insult associated with cisplatin-induced hepatic and renal toxicity, co-administering antioxidantagents is urgently needed to minimize potential side effects through inhibition of cisplatin-induced free radical generation [6].

Plant-based antioxidants with diverse pharmacological actions have come under extensive study. Asteraceae is one of the largest plant families, with above 1620 genera and 23,600 species around the world. Family Asteraceae demonstrated remarkable antioxidant, anti-inflammatory, antimicrobial, wound healing, and cytotoxic activities. Those beneficial pharmacological effects may be attributed to the chemical diversity of different metabolites such as polyphenols, phenolic acids, flavonoids, polysaccharides, and terpenes [7]. Genus *Sonchus* (Family: Asteraceae) is a class of edible wild plants that are widely spread in Europe, Asia, and Africa. The *Sonchus* species possess hepatoprotective, cardioprotective, anticancer, antioxidant, anti-inflammatory, and antimicrobial effects. Numerous bioactive metabolites (e.g., fatty acids, tocopherols, flavonoids, triterpenes, coumarins, steroids, and saccharides) were detected in *Sonchus* species [8]. *Sonchus cornutus* is an annual herb with hollow stem, sessile leaves, and large inflorescence. *S. cornutus* is commonly known as *Molieta* in Sudan and traditionally used for treatment of diabetes, hypertension, and malaria. The parasitic activity of *S. cornutus* against *Theileria lestoquardi* was previously observed. Previous studies have reported that the therapeutic dose (50–500 mg/Kg) of *S. cornutus* was safe and caused no hematological, biochemical, or histopathological harmful effects [9]. Despite the biological value and safety of edible *S. cornutus*, few chemical studies were performed.

Therefore, the present study was directed to detect the secondary metabolites in *S. cornutus* extract using liquid chromatography-tandem mass spectrometry (LC-MS/MS), analyze major chemical composition, and evaluate its in vitro antioxidant activity. The current study also evaluated the protective effect of *S. cornutus* crude extract against cisplatin-induced hepato- and nephro-toxicities in experimental mice.

## 2. Materials and Methods

### 2.1. Collection of Plant Material and Extraction Process

The whole plant was collected in November from Khartoum, Sudan. *S. cornutus* was air dried, finely ground, and then was subjected to cold maceration with methyl alcohol (70%; 3 × 1000 mL) at room temperature. The combined hydroalcoholic solutions were concentrated under vacuum to give a crude extract of 75 gm weight.

### 2.2. Metabolic Profiling and Chemical Analysis of S. cornutus

#### 2.2.1. LC-MS/MS Metabolic Profiling

The metabolic profiling and composition were established using (HPLC/Triple-TOF-MS/MS) as mentioned before [10,11,12,13,14]. Detailed methodology is provided in Appendix A.

#### 2.2.2. Total Phenolics Content Assay

Folin–Ciocalteu (Folin-C) spectrophotometric method was applied for determination of total polyphenolics content. Briefly, an amount of 500 µL of crude extract of *S. cornutus* was mixed with 2.5 mL of Folin-C reagent (10%) and then 2 mL of sodium carbonate (7.5%) was added after 5 min. The reaction was allowed to complete for 2 h in darkness at 25 °C [15]. The intensity of the resulting blue color was measured spectrophotometrically at 760 nm using distilled water as a blank. The experiment was replicated three times. The total phenolic content of *S. cornutus* was calculated in comparison with a calibration curve using gallic acid as a standard; therefore, the values are expressed as gallic acid equivalent per gram extract (mg GAE/gm).

#### 2.2.3. Total Flavonoids Content Assay

Aluminum complexation reaction was applied for determination of total flavonoids content. A volume of 1 mL of *S. cornutus* sample was mixed with 4 mL of distilled water and 0.3 mL of sodium nitrite solution (5%). After 5 min, 0.3 mL of AlCl_3_. 6H_2_O solution (10%) was added and the mixture was incubated for 5 min at 25 °C. Finally, 2 mL of 1 M NaOH solution was added, and the total volume was completed to 10 mL with distilled water. The intensity of yellowish orange color was recorded spectrophotometrically at 510 nm using distilled water as a blank [16]. Additionally, the experiment was repeated three times. The total flavonoids content of *S. cornutus* was calculated in comparison with a calibration curve using quercetin as a standard; therefore, the values are expressed as quercetin equivalent per gram extract (mg QE/gm).

### 2.3. In Vitro Antioxidant Assays

#### 2.3.1. Determination of Total Antioxidant Capacity

Phosphomolybdenum assay was conducted for evaluation of total antioxidant capacity of *S. cornutus*. This method was carried out as follows: A volume of 0.3 mL of methanolic crude extract of *S. cornutus* was mixed with 2.7 mL of phosphomolybdenum reagent solution. The reagent solution was prepared by mixing 28 mM sodium phosphate and 4 mM ammonium molybdate in 0.6 M sulfuric acid. Then, the sample–reagent mixture was capped and incubated for 90 min at 95 °C. After the reaction time, the mixture was cooled for 20 min at ambient temperature and then the intensity of green color was recorded spectrophotometrically at 695 nm using methanol as a blank [17]. The experiment was done in triplicate and a calibration curve using gallic acid was produced; the values were expressed as gallic acid equivalent per gram extract (mg GAE/gm). Ascorbic acid was used as a reference control.

#### 2.3.2. Ferric Reducing Antioxidant Power Assay

The reducing potential of *S. cornutus* was performed through mixing 2 mL of crude extract, 2 mL of sodium phosphate buffer solution (0.2 M; pH = 6.6), and 2 mL of potassium ferricyanide (10 mg/L). The reaction mixture was incubated for 20 min at 50 °C. Then, the mixture was acidified using 2 mL of trichloroacetic acid (100 mg/L) and centrifuged. A volume of 2 mL of the supernatant was collected, diluted with 2 mL of distilled water and 0.4 mL of ferric chloride (0.1%), and then incubated for 10 min. The intensity of the color was measured spectrophotometrically at 700 nm using methanol as a blank [18]. The experiment was also done in triplicate. The reducing potential of *S. cornutus* was calculated in comparison with a calibration curve using 1 mM ferrous sulphate; therefore, the values are expressed as mM ferrous equivalent per gram extract (mmol Fe^2+^/gm) [19]. Ascorbic acid was applied as a reference standard.

#### 2.3.3. DPPH Radical Scavenging Assay

The free radical scavenging activity of *S. cornutus* was evaluated via mixing equal volumes of the crude extract with 1,1-diphenyl-2-picrylhydrazyl (DPPH) free radical (200 μM). The reaction mixture was incubated for 30 min in darkness at 30 °C. The intensity of yellow color was measured spectrophotometrically at 516 nm using methanol as a blank. The experiment was also done in triplicate. The scavenging percentage was calculated based on the following equation: [(A_control_ − A_sample_)/A_control_ × 100], where A _control_ is the absorbance of the DPPH solution only, and A_sample_ is the absorbance of both DPPH solution and the sample at different concentrations (5–1000 µg/mL) [17]. The value of IC_50_ (the concentration that is required to inhibit initial concentration of DPPH by 50%) was determined by a linear concentration/percentage inhibition curve. Ascorbic acid was also applied as a reference standard.

### 2.4. In Vivo Study of S. cornutus

#### 2.4.1. Drugs and Chemicals

Cisplatin (50 mg/50 mL) vial was purchased from Mylan Company (Saint Priest, France). Phosphate buffered saline solution (PBS) was obtained from Lonza Bio-products (Verviers, Belgium).

#### 2.4.2. Experimental Animals and Study Protocol

Experimental design and animal handling were carried out in accordance with Guide for the Care and Use of Laboratory Animals (8th edition, National Academies Press) and were approved by the Ethics Committee of Faculty of Pharmacy, Suez Canal University (Ethics code: 202103RA2).

Thirty-two male Swiss albino mice with an average weight of 25 ± 5 g, obtained from the Faculty of Veterinary Medicine, Suez Canal University, Ismailia, Egypt, were utilized in this study. Mice were accommodated in stainless steel rodent cages under controlled room temperature (25 ± 2 °C) with a 12 h light–dark cycle. Mice were kept on standard rat chow and tap water. They were kept for a one-week adaptation period. After the acclimatization period, mice were randomly divided into four groups (eight animals each). The first and second groups received saline orally once daily for 3 weeks, and on the 15th day of the experiment, received a single intraperitoneal (i.p.) injection of isotonic saline or cisplatin (7.5 mg/kg) [20], serving as the normal control and cisplatin groups, respectively. The third and fourth groups received *S. cornutus* extract (250 mg/kg/day; orally) and (500 mg/kg/day; orally), respectively, for 3 weeks, starting 14 days before giving a single i.p. injection of cisplatin (7.5 mg/kg).

At the end of the experiment, the animals were overnight-fasted, and blood samples were collected from the retro-orbital plexus under mild ketamine anesthesia. Samples were left to clot and then centrifuged for 20 min at (2000× *g*/4 °C) for serum separation. The obtained sera were kept at −80 °C until used to assess alanine aminotransferase (ALT), aspartate aminotransferase (AST), and alkaline phosphatase (ALP) activities, total protein, albumin, blood urea nitrogen (BUN), and serum creatinine. Later, mice were sacrificed by decapitation and their mice liver and kidneys were dissected out, weighed, and washed with saline, then divided into three portions: one portion was fixed in 10% buffered formalin. After that, the formalin-fixed samples were washed with tap water, immersed in ethyl alcohol serial dilutions, embedded in paraffin, and cut into sections (4–5 μm thickness) using a rotatory microtome for histopathological and immunohistochemical analysis. The second portion was homogenized in phosphate buffer (pH 7.4), then centrifuged at (2000× *g*/4 °C) for 15 min. The resulting supernatant was stored at −80 °C for further biochemical analysis. The third portion was homogenized in Qiazol reagent and used for quantitative PCR analysis.

### 2.5. Biochemical Estimations and Histopathological Examination

#### 2.5.1. Assessment of Liver and Kidney Functions

Serum levels of ALT, AST, ALP, creatinine, and BUN were measured by routine colorimetric methods using commercial diagnostic kits (Diamond Diagnostics, Cairo, Egypt) according to the manufacturer’s data protocols.

#### 2.5.2. Assessment of Hepatic and Renal Oxidative Stress and Inflammatory Markers

In the liver and kidney homogenates of all groups, malondialdehyde (MDA) levels and catalase and SOD activities were measured using colorimetric assay kits obtained from Biodiagnostic Co. (Dokki, Giza, Egypt; Cat. No. MD 2529 for MDA, CA 2517 for catalase and SD 2521 for SOD) in accordance with manufacturer’s instructions.

Levels of ROS, reduced glutathione (GSH), and oxidized glutathione (GSSG) were determined in the tissue homogenates of both liver and kidney by mouse-specific enzyme-linked immunosorbent assay (ELISA) kits (MyBioSource, San Diego, CA, USA; Cat. No. MBS2601061, MBS724815, and MBS749109, respectively). Levels of GSH and GSSG were used to calculate the GSH/GSSG ratio in the liver and the kidney of the experimental mice.

Nitric oxide (NO) levels were determined in tissue homogenate spectrophotometrically (Biodiagnostic, Egypt; Cat. No. NO 2533). Hepatic and renal iNOS was determined using mouse-specific ELISA kits (MyBioSource, San Diego, CA, USA; Cat. No. MBS263618).

Interleukin (IL)-6, IL-1β, and IL-10 were measured using mouse-specific ELISA kits purchased from MyBioSource (San Diego, CA, USA; Cat. No. MBS730957, MBS701092, and MBS018124, respectively), following manufacturer’s instructions.

#### 2.5.3. Quantitative Real-Time—PCR Analysis of Nuclear Factor kappa B (NF-κB), Tumor Necrosis Factor-α (TNF-α), and Nuclear Factor Erythroid 2-related Factor 2 (Nrf2) Expression

Total RNA was extracted from hepatic and renal tissues using miRNeasy Mini Isolation kit (Qiagen, Hilden, Germany) according to the manufacturer’s protocol and quantified spectrophotometrically using the NanoDrop ND-1000 (NanoDrop Tech., Wilmington, DE, USA) at 260/280 nm. NF-κB, TNF-α, and Nrf2 expressions were assayed using GoTaq^®^ 1-Step RT-qPCR System (Promega, Madison, USA) with the target genes’ expression compared to β-actin mRNA gene expression as a reference control. According to the manufacturer’s instructions, the total volume of the reaction was 20 μL containing 10 μL of GoTaq^®^ qPCR master mix, 0.4 μL of GoScript™ RT mix for 1-step RT-qPCR, 2 μL of forward and reverse primers, 0.31 μL of CXR reference dye, 4 μL of RNA template and 3.29 μL of nuclease-free water. The GenBank accession No., primers, and annealing temperatures are listed in Appendix A.

The thermal cycler conditions were 15 min at 37 °C (reverse transcription), 10 min at 95 °C (reverse transcriptase inactivation), 40 cycles of 10 s at 95 °C (denaturation), 30 s at the specified annealing temperature (annealing), 30 s at 72 °C (extension), and one cycle for 10 min at 72 °C (final extension). The reaction for each sample was run in duplicate, and a no-template control was run for all samples. The fold change was determined according to the comparative cycle threshold method (ΔΔCT method) [21].

#### 2.5.4. Immunohistochemistry for Detection of Bax, Bcl-2, and Caspase-3

Sections of the liver and kidney tissues were deparaffinized and dehydrated sequentially in graded ethyl alcohol. After that, sections were autoclaved at 121 °C in distilled water for 5 min to unmask antigen sites. Then, the slides were immersed in 3% H_2_O_2_ for quenching the endogenous peroxidase activity and blocked in 5% bovine serum albumin blocking reagent for 20 min after rinsing 3 times in PBS to stop non-specific reactions. The blocking slides were incubated overnight at 4 °C with anti-Bax, anti-Bcl-2, and anti-caspase 3 antibodies (Invitrogen Life technologies, Carlsbad, CA, USA, Cat. No. 33-6400, MA5-15668, and 43-7800, respectively) diluted in PBS, followed by secondary antibody incubation for 30 min. Bound antibodies were detected by incubation for 45 min with avidin–biotin complex (ABC kit, Vector Laboratories, Burlingame, CA, USA) at 37 °C. Visualization of the reaction product was achieved by treatment with diaminobenzidine substrate (Sigma Chemical Company, St. Louis, MO, USA) and counterstaining with Mayer’s hematoxylin. Proteins’ expression was observed under light microscopy (Olympus BX-60, Tokyo, Japan). Quantification of positive areas was carried out using image analysis system Leica Qwin DW3000 (LEICA Imaging Systems Ltd., Cambridge, UK). Five fields/each slide were measured in all groups. The positive staining intensity was measured as the ratio of the stained area to the entire field.

#### 2.5.5. Histopathological Examination

The cut sections obtained from the prepared blocks were stained with hematoxylin and eosin (H&E) and visualized under a light microscope for examination of hepatic and renal histological structure.

#### 2.5.6. Statistical Analysis

Statistical analysis was performed using the Statistical Package for Social Sciences, SPSS (IBM, Armonk, NY, USA), v. 21.0 software. Values were expressed as the mean ± standard deviation (SD). One-way analysis of variance (ANOVA) followed by Bonferroni’s post hoc test was used for statistical comparisons. A statistically significant difference was indicated by a *p* value less than 0.05.

## 3. Results and Discussion

### 3.1. LC-MS/MS Metabolic Profiling of Crude Extract of S. cornutus

Liquid chromatography-tandem mass spectrometry (LC-MS/MS) is an informative bioanalytical chromatographic tool based on physical separation of metabolites of different classes followed by mass fragmentation-based identification. The tandem mass spectrometry and mass error accuracy are sufficient interpretative parameters for identification of deduced compounds [10,11]. The current LC-MS/MS study of the crude extract of *S. cornutus* was developed for the first time. The metabolomic analysis (Appendix A) revealed the prevalence of polyhydroxylated phenolic compounds such as anthocyanidin glycosides, coumarins and their glycosides, flavonoids and their glycosides, and phenolic acids (Table 1, Figure 1).

Polyphenols are well-known for their antioxidant and anti-carcinogenic activities, so they draw great attention of scientific research for health care. The detected metabolites were previously reported to play a crucial role in prevention and treatment of several diseases. Briefly, cyanidin-3-O-rutinoside inhibited α-glucosidase enzyme, resulting in lowering postprandial hyperglycemia in diabetic rats [30]. Delphindin-3-glucoside suppressed the expression of HOTAIR, a diagnostic biomarker for several cancer types [31]. Furthermore, the in silico study of peonidin-3-O-glucoside demonstrated anti-inflammatory activity through binding to TNF-α receptor and subsequently inhibiting TNF-α signaling [32]. Additionally, cyanidin-3-O-glucoside exhibited a protective mechanism against intestinal injury through its antioxidant and anti-inflammatory properties. These biological activities were declared through upregulation of glutathione and downregulation of both cyclooxygenase-2 and iNOS. Additionally, the cytotoxic potential of cyanidin-3-O-glucoside referred to angiogenesis suppression, apoptosis induction, and downregulation of proapoptotic factors as Bax and caspases [33]. Concerning the detected coumarins and their glycosides, esculetin showed antiviral activity against human immunodeficiency (HIV) virus [34], while esculin showed significant antimicrobial activity against Gram-positive bacteria [35]. Daphnetins hold anti-inflammatory, antithrombotic, immunosuppressive, and anti-*Helicobacter pylori* activities [36,37,38]. Additionally, fraxetin exhibited antibacterial activity against *Staphylococcus aureus* [39] and induced apoptosis in colon cancer cells [40]. Regarding the detected flavonoids and their glycosides, orientin was found to be a promising antimicrobial, anti-inflammatory, cardioprotective, and anti-obesity agent [41]. The anti-inflammatory effect of luteolin was discussed through suppression of cyclooxygenase-2 [42] while fisetin suppressed mast cells activation which are involved in inflammatory response [43]. Referring to phenolic acids, caffeic acid exhibited innumerable physiological activities such as antioxidant, anti-inflammatory, antibacterial, immunomodulatory, hepatoprotective, anticancer, and anti-atherosclerotic activities [44]. Rosmarinic acid is a promising candidate for treatment of liver fibrosis through suppression of pro-inflammatory and fibrotic markers [45]. Chlorogenic acid is a potent antimicrobial, anti-obesity, anti-inflammatory, hepatoprotective, neuroprotective, and cardioprotective medicinal agent [46]. Ursolic acid was also shown to be an anti-inflammatory, antitumor, and antimicrobial agent [47]. The previously reported antioxidant properties of different species of genus *Sonchus* were correlated to the polyphenolic active metabolites [48,49,50]. Therefore, quantitative analysis of polyphenols in *S. cornutus* was a must for our study.

### 3.2. Quantification of Total Phenolics and Total Flavonoids in S. cornutus

Phenolic compounds are widely spread secondary metabolites in plants. Polyphenols exhibit numerous pharmacological health benefits as they act as antioxidants through scavenging free radicals [51]. Their antioxidant effect defends against inflammation and oxidative stress-induced diseases, cancer, diabetes, aging, pulmonary, cardiovascular, and neurological disorders [52]. Based on the predominance of polyphenolic compounds in the above-mentioned LC-MS/MS analysis, there was a deep need to quantify both total phenolics and total flavonoids in *S. cornutus*. From data shown in Table 2, it was found that the crude extract of *S. cornutus* is an enriched source of polyphenols which were determined to be 206.28 ± 14.64 mg gallic acid equivalent/gm extract for total phenolics and 45.56 ± 1.78 mg quercetin equivalent/gm extract for total flavonoids.

These findings are in accordance with those found in *Sonchus asper*, in which total phenolics and total flavonoids were found at concentrations of 332 ± 1.53 mg gallic acid equivalent/gm dry extract and 11.4 ± 0.45 mg rutin equivalent/gm dry extract, respectively [53]. It can be declared that *Sonchus* species are rich in flavonoids and other polyphenolic compounds such as tannins [54], coumarins, and phenolic acids [55].

### 3.3. In Vitro Antioxidant Activity of S. cornutus

The polyphenolic compounds are renowned for their antioxidant activity. Therefore, the in vitro antioxidant activity of *S. cornutus* was determined for the first time by using three different methods: total antioxidant capacity, ferric reducing power capacity, and DPPH free radical scavenging activity assays. Among them, the DPPH scavenging activity test illustrates the ability of *S. cornutus* to donate proton to the violet solution of DPPH forming stable yellow solution of 1,1-diphenyl-2-picrylhydrazine [56]. Unlike DPPH assay, total antioxidant capacity and reducing power activity are based on the ability of *S. cornutus* to reduce molybdenum VI to molybdenum V or ferric to ferrous, respectively [56]. From data shown in Table 3, the crude extract of *S. cornutus* showed a significant DPPH free radical scavenging activity with IC_50_ of 16.10 ± 2.14 µg/mL when compared to ascorbic acid (10.64 ± 0.82 µg/mL) as a reference standard. This can be explained by *Sonchus* species’ richness of polyphenolic compounds [54] that can easily donate proton to stabilize DPPH free radical.

Moreover, the crude extract of *S. cornutus* exhibited moderate reducing potential that was evaluated via total antioxidant capacity and ferric reducing power assays. The ferric reducing power activity of *S. cornutus* was 1.92 ± 0.71 mM Fe^+2^/gm while the total antioxidant capacity was 49.06 ± 3.62 mg GAE/gm (Table 3). Based on the hopeful in vitro antioxidant effect of the crude extract of *S. cornutus*, further in vivo assessment of hepatoprotective and nephroprotective effects of *S. cornutus* against cisplatin-induced toxicity was held.

### 3.4. The Effects of the Crude Extract of S. cornutus against Cisplatin-Induced Hepatic and Renal Toxicity in Experimental Mice

Investigation of the liver and kidney functions in the sera of the experimental mice revealed that administration of cisplatin (7.5 mg/kg) alone caused a significant increase in the serum levels of ALT, AST, ALP, creatinine, and BUN relative to the normal control group (*p <* 0.05). Mice that received both doses of the crude extract of *S. cornutus* (250 and 500 mg/kg) before a single dose (7.5 mg/kg) of cisplatin had significantly lower levels of all those markers compared to the cisplatin control group. The levels of all the detected liver and kidney markers were significantly lower in the group that received higher dose (500 mg/kg) of *S. cornutus* extract compared to the group that received the lower dose (250 mg/kg) (Table 4).

The antioxidant and anti-inflammtory properties of the crude extract of *S. cornutus* in the cisplatin receiving mice were assessed. As expected, cisplatin caused a significant increase of the levels of MDA, ROS, NO, and iNOS, and a significant decrease of GSH/GSSG ratio, SOD, and catalse in both the liver and kidney tissues compared to the normal control group (*p <* 0.05) (Table 5).

Both doses (250 and 500 mg/kg) of *S. cornutus* significantly decreased the levels of MDA, ROS, GSSG, NO, and iNOS, and increased concentrations of GSH, SOD, and catalase in the liver tissue in comparison with the cisplatin control group. The higher dose of *S. cornutus* (500 mg/kg) significantly decreased levels of MDA and iNOS and increased the ratio of GSH/GSSG and the activity of catalase in the liver tissue relative to the *S. cornutus* (250 mg/kg) group (Table 5).

In the renal tissue, both doses of *S. cornutus* crude extract lowered the GSH/GSSG ratio and decreased the levels of MDA, ROS, and iNOS compared to the cisplatin control group, however, only mice that received 500 mg/kg of *S. cornutus* extract had significantly higher SOD and catalase and lower NO in the kidney tissue relative to the cisplatin control group. The higher dose of *S. cornutus* (500 mg/kg) produced a more significant lowering of MDA, ROS, and the GSH/GSSG ratio compared to the 250 mg/kg *S. cornutus* dose (Table 5).

Interestingly, catalse levels in both the liver and renal tissues of the *S. cornutus* (500 mg/kg) group were not significantly different compared to the normal control group (Table 5).

Additionally, the expression of NF-κβ and TNF-α and the protein levels of IL-1β and IL-6 were significantly higher in both the liver and kidney tissues of the mice that received cispaltin (7.5 mg/kg) only compared to the normal control group, whereas the expression of Nrf2 and the levels of IL-10 were significantly lower in both tissues of the cisplatin control group relative to the normal mice (*p <* 0.05). Administration of either doses (250 mg/kg and 500 mg/kg) of *S. cornutus* crude extract significantly decreased the expression of NF-κβ and TNF-α, as well as the levels of IL-1β and IL-6, but significantly increased the expression of Nrf2 and the levels of IL-10 in both the liver and kidney tissues compared to the cisplatin control group (Figure 2 and Figure 3).

In the liver tissue, the group that was given 500 mg/kg of *S. cornutus* had significantly lower NF-κβ expression and lower IL-1β and IL-6 levels, but significantly higher Nrf2 expression and IL-10 levels compared to the *S. cornutus* (250 mg/kg) group. Expression of TNF-α in the liver of *S. cornutus* (500 mg/kg) group showed no significant difference relative to the normal control group (Figure 2).

In the renal tissue, the *S. cornutus* (500 mg/kg) receiving mice had significantly lower IL-6 and siginificantly higher IL-10 levels compared to the *S. cornutus* (250 mg/kg) group. Interestingly, the exression of NF-κβ in the kidey tissue of the group that received 500 mg/kg of *S. cornutus* crude extract was not significantly different compared to the normal control group. Moroever, the expression levels of TNF-α and Nrf2 in both groups that received the lower and higher doses of *S. cornutus* were not significantly different relative to the normal control group (Figure 3).

The apoptotic effect of cisplatin was evidenced by the significantly increased expression of caspase-3 (Figure 4) and Bax (Figure 5), and decreased expression of Bcl-2 (Figure 6) in both the liver and kidney tissues of the cisplatin group in comparison to the normal control group. Immunohistochemical assessement showed that administration of 250 mg/kg of the crude extract of *S. cornutus* significantly downregulated caspase-3 (Figure 4) and upregulated Bcl-2 expression (Figure 6) in the liver tissue only relative to the cisplatin control group. The higher dose (500 mg/kg) of *S. cornutus* significantly downregulated caspase-3 (Figure 4) and Bax (Figure 5), but upregulated Bcl-2 (Figure 6) in both the liver and kidney tissues in comparison to the group that was given cisplatin only. The renal expression of caspaase-3 (Figure 4) and Bax (Figure 5) in the *S. cornutus* (500 mg/kg) group was significantly lower than their expression in the *S. cornutus* (250 mg/kg) group. Moroever, the expression of Bcl-2 in both the liver and the kidney of the mice that received 500 mg/kg of *S. cornutus* was significantly higher than its expression in the lower dose (*S. cornutus* 250 mg/kg) group (Figure 6). Notably, the hepatic expression of caspase-3 (Figure 4) and Bcl-2 (Figure 6) in the higher dose (*S. cornutus* 500 mg/kg) group was not significantly different relative to the normal control group.

Additionally, histopathological examination of the liver tissue revealed normal architecture of the liver in the normal control group (Figure 7A), while the liver specimens of mice that were given cisplatin alone showed inflammation, foci of spotty necrosis, with congestion of central vein and dilation of sinusoidal spaces (Figure 7B). The group that received 250 mg/kg of *S. cornutus* crude extract + cisplatin showed partial improvement of the liver tissue, less infiltration of inflammatory cells, but dilated sinusoidal spaces were still detected (Figure 7C). The liver tissue of the *S. cornutus* (500 mg/kg) group showed the best improvement, with minimal infiltration of inflammatory cells and restoration of normal sinusoidal spaces (Figure 7D).

On the other hand, histopathological examination of the kidney tissue revealed normal looking glomeruli, proximal tubules, and distal tubules in the normal control group (Figure 8A). The renal injury was pronounced in the group that received cisplatin only, where patchy interstitial inflammatory cell infiltrate, congestion, focal tubular cell necrosis with condensed eosinophilic cytoplasm, and regenerative changes were detected (Figure 8B). No significant improvement was observed on examination of the kidney tissue of the mice that received the lower dose of *S. cornutus* (250 mg/kg) + cisplatin (Figure 8C), however, the higher dose (500 mg/kg) of *S. cornutus* crude extract showed improved renal tissue, with patchy interstitial inflammatory cell infiltrate, mild congestion, and minimal focal tubular cell necrosis (Figure 8D).

The hepatotoxicity and nephrotoxicity induced by cisplatin are well-documented and experimental models that have been extensively used to evaluate the protective effects of various reagents against cisplatin-induced hepato- and nephrotoxicity. Liver is the major organ in which most metabolic reactions occur. Cisplatin is metabolized in the liver by cytochrome P450 (CYP450) enzyme complex [57], where cytochrome P450 2E1 (CYP2E1) is thought to be the leading enzyme involved in cisplatin-induced hepatotoxicity [58]. Cisplatin-treated cells overexpressing CYP2E1 were proven to produce high levels of ROS [59]. In addition, cisplatin has been reported to cause direct damage to mitochondrial DNA, leading to reduction of mitochondrial protein synthesis and impairment of electron transport chain function. Such events are associated with increased intracellular ROS levels [60]. ROS play a key role in cisplatin-induced hepatotoxicity through increasing hepatic MDA [61]. The latter is the principal product of polyunsaturated fatty acid peroxidation and is considered a biomarker of lipid peroxidation and oxidative stress. MDA can react with cellular proteins and DNA, resulting in biomolecular damage [62]. It has been shown that a set of endogenous antioxidants can reduce ROS-induced cellular damage. SOD generally dismutases the superoxide anion radical into hydrogen peroxide which is degraded by catalase or glutathione peroxidase using GSH [63]. However, increased oxidative stress leads to exhaustion of GSH and inactivation of antioxidant enzymes such as SOD and catalase either by cross linking with MDA or rapid consumption of the antioxidant molecules in fighting the generated free radicals, further aggravating oxidative stress [61,64].

Moreover, the accumulation of intracellular ROS is closely related to the induction of pro-inflammatory cytokines via NF-ĸB and various pro-apoptotic cellular signaling cascades [65]. NF-κB has been reported to affect downstream target genes’ expression through downregulating Nrf2 transcription and activity by competing for its transcriptional co-activator cAMP response element binding protein [66]. Interestingly, Nrf2 has been proved to maintain intracellular redox homeostasis and exert anti-inflammatory functions through regulating heme oxygenase-1 (HO-1) axis and counteracting NF-κB driven inflammatory response [67]. Cisplatin was previously reported to increase the expression of pro-inflammatory cytokines such as TNF-α, IL-1β, and IL-6 [68,69], along with decreasing levels of the anti-inflammatory IL-10 in the liver tissue [70]. Additionally, increased levels of NO and iNOS, key mediators of both oxidative stress and inflammation, were reported in cisplatin administered rodents [71]. Moreover, cisplatin-induced oxidative stress and inflammation have been reported to produce hepatocyte apoptosis through activation of executioner caspase-3 that triggers release of cellular proteins and DNA fragmentation factors, leading to characteristic changes of apoptosis. Caspase-3 activation is mainly regulated by the Bcl-2 family proteins through activation of the pro-apoptotic protein Bax that in turn inactivates anti-apoptotic protein Bcl-2. Bax triggers a sequence of events that cause alterations in mitochondrial permeability and stimulation of cytochrome c release with subsequent activation of caspases and cell death. Such events are inhibited by Bcl-2 [72]. Cisplatin-induced hepatotoxicity was proven by increased serum levels of the hepatic enzymes ALT, AST, and ALP [73].

More to the point, cisplatin is cleared by the kidneys via both glomerular filtration and tubular secretion, and it accumulates in the renal cells in concentrations higher than those in the blood [74]. Increased serum levels of creatinine and urea were detected in cisplatin-receiving rats [75]. Cisplatin is suggested to undergo metabolic activation in proximal tubular cells to highly reactive and toxic thiols, mainly through its conjugation with GSH, leading to exhaustion of the latter [76]. Production of ROS, depletion of antioxidant enzymes, and accumulation of the lipid peroxidation products are among the main mechanisms involved in cisplatin-induced nephrotoxicity [77]. Excess production of NO by iNOS enzyme was also detected following cisplatin administration [78]. Additionally, cisplatin-induced nephrotoxicity has been reported to involve nuclear and mitochondrial DNA injury, along with the induction of apoptotic and inflammatory pathways [79]. Moreover, increased NF-ĸB and TNF-α expression and decreased Nrf2 expression have been shown to be associated with cisplatin-induced nephrotoxicity, leading to enhanced levels of inflammatory cytokines [80,81,82]. Furthermore, increasing lines of evidence show that apoptosis is implicated in cisplatin-associated nephrotoxicity through the imbalance between the pro-apoptotic protein Bax and the anti-apoptotic protein Bcl-2, triggering cytochrome c release and caspase-3 activation [83].

A wide variety of natural reagents (i.e., plant extracts, flavonoids, phenolic compounds, terpenoids, and oils) were reported to protect against cisplatin-induced hepatotoxicity through their antioxidant, anti-inflammatory, and antiapoptotic properties [73]. Moreover, numerous natural products (flavonoids, polysaccharides, phenylpropanoids, saponins, etc.) have been shown in recent in vivo and in vitro studies to antagonize the mechanisms involved in cisplatin-induced kidney damage through their antioxidant, anti-inflammatory, and anti-apoptotic activities [84].

Antioxidant properties of *Sonchus* species were previously described. Examples are *S. oleraceus* [48], *S. arvensis* [49], and *S. asper* [50]. Interestingly, *S. asper* methanolic extract was reported to alleviate oxidative stress, decrease lipid peroxidation, and enhance antioxidant defenses in a model of CCl_4_-induced nephrotoxicity [85]. Moreover, the methanolic extract of *S. asper* produced similar protective effects against CCl_4_-induced oxidative damage in the liver [86]. The extract of *S. oleraceus* was also found to protect against kidney injury induced by ischemia-reperfusion in rats and to restore the levels of creatinine, BUN, MDA, SOD, and proinflammatory cytokines [87].

There is a deep correlation between the high content of total phenolic compounds in *S. cornutus* crude extract and its observed antioxidant and anti-inflammatory effects in the current work. The antioxidant capacity of polyphenols is based on their ability to inhibit enzymes involved in the production of ROS, in addition to their ROS scavenging properties, and the upregulation of the antioxidant defenses such as GSH, SOD, and catalase [88]. Polyphenols are also able to affect the production of the key inflammatory mediator NO by interfering with the activity of iNOS enzyme [89]. Considering that oxidative stress induces a variety of transcription factors, leading to increased expression of proteins involved in inflammatory pathways such as TNF-α and IL-6, the anti-inflammatory effect of polyphenols could be attributed to their radical scavenging activities and modulation of NF-ĸB and mitogen-activated protein kinase (MAPK) signaling pathways [90,91]. Modulation of such pathways contributes to transcriptional upregulation and increased nuclear translocation of Nrf2 consequently leading to enhanced expression of Nrf2 downstream genes, including antioxidant and phase II detoxification enzymes [92].

In agreement with the findings of the current study, some of the metabolites detected in the extract of *S. cornutus* by LC-MS/MS exhibited hepato- and nephroprotective effects against cisplatin-induced toxicity. For instance, rosmarinic acid was reported to protect against cisplatin-induced kidney injury in mice through amelioration of oxidative stress, reduction of the apoptotic markers p53 and phosphorylated p53, and inhibition of the expression of caspase-3, NF-ĸB, and TNF-α [93]. Chlorogenic acid also decreased serum creatinine and BUN levels, reduced oxidative stress and TNF-α expression, and suppressed p53, Bax, and active caspase-3 in a murine model of cisplatin-induced nephrotoxicity [94]. Additionally, it has been demonstrated that antioxidant and anti-inflammatory effects of chlorogenic acid are attributable to enhanced Nrf2 nuclear translocation and HO-1 expression and reduced IĸB phosphorylation and subsequent NF-ĸB nuclear translocation [95]. Urosalic acid was reported to protect against cisplatin-induced nephrotoxicity through its antioxidant and anti-apoptotic activities and mitigation of the inflammatory markers IL-1β, IL-6, and TNF-α [96].

Moreover, caffeic acid has been shown to abrogate hepatorenal toxicities mediated by different toxicants by virtue of its antioxidant, anti-inflammatory, and anti-apoptotic effects [97]. It is well-established that caffeic acid effectively quenches the free radicals and acts as a chain breaking antioxidant, terminating the chain reaction of lipid peroxidation and minimizing its detrimental effects. Additionally, it was found to increase the activities of antioxidant enzymes, including SOD and catalase [98]. Additionally, antioxidant activity of caffeic acid has been reported to be attributed to its ability to phosphorylate kelch-like ECH-associated protein-1(Keap1), leading to its degradation, thus releasing Nrf2 from the Keap1-mediated inhibitory complex, further contributing to maintenance of the cellular antioxidant defense system [97]. In addition, caffeic acid has been demonstrated to inhibit NF-kβ activation by inhibiting Ikβ kinase (IKK) complex activity needed to degrade Ikβα, which is responsible for sequestering NF-kβ in the cytoplasm, thereby mitigating pro-inflammatory cytokines expression including IL-1β and TNF-α. Importantly, caffeic acid anti-apoptotic activity has been shown to be attributed to its ability to resolve oxidative stress and inflammation, as well as repressing Bax activity, with the simultaneous increase in Bcl-2 level leading to decreased caspase-3 activity [97].

The bioactive flavonoid luteolin ameliorated oxidative/nitrosative stress, suppressed the expression of the inflammatory factors NF-ĸB and TNF-α, and downregulated the renal apoptotic markers p53 and caspase-3 in a mouse model of cisplatin-induced nephrotoxicity [99]. Similarly, the flavonoid fisetin restored oxidative balance, blocked the activation of NF-ĸB signaling, and reduced the levels of MDA, TNF-α, and iNOS in a rat model of cisplatin-induced nephrotoxicity. The antioxidant and anti-inflammatory effects of fisetin were accompanied by anti-apoptotic effects, where it decreased the renal expression of the pro-apoptotic proteins Bax, cleaved caspase-3, cleaved caspase-9 and p53, and upregulated the anti-apoptotic protein Bcl-2 [100]. The coumarin daphnetin was also shown to reduce levels of creatinine and BUN, decrease renal MDA, inhibit NF-ĸB, and downregulate TNF-α and IL-1β in a model of cisplatin-induced nephrotoxicity [101]. In a rat model of cisplatin-induced hepatotoxicity, fraxetin, a bioactive coumarin, enhanced the antioxidant activity and decreased the expression of caspase-3 and TNF-α in the liver [102].

## 4. Conclusions

In conclusion, the LC-MS/MS metabolic analysis profile revealed enrichment of *S. cornutus (Asteraceae)* with polyphenols such as anthocyanidin glycosides, coumarins, flavonoids and their respective glycosides, and phenolic acids. In vitro assays revealed that the crude extract of *S. cornutus* exhibited a significant DPPH free radical scavenging effect and a promising reducing potential, indicated by the total antioxidant capacity and ferric reducing power capacity assays. Besides, this study was the first to evaluate hepato- and nephroprotective effects of *S. cornutus* extract using a mouse model of cisplatin-induced hepato- and nephrotoxicity. As per our findings, it has been clearly indicated that pretreatment with *S. cornutus* extract for 14 days before cisplatin and 7 days after cisplatin injection improved liver and kidney functions, restored oxidative stress, inflammatory, and apoptotic markers. Based on these observations, the antioxidant effects of *S. cornutus* extract against cisplatin-induced liver and kidney injuries were evidenced by its ability to scavenge free radicals, reduce oxidative stress, and elevate the activities and levels of antioxidant molecules. In addition, administration of *S. cornutus* extract effectively inhibited NF-kβ signaling pathways with subsequent decrease in the pro-inflammatory cytokines’ expression including TNF-α, IL-1β and IL-6 and elevation of the anti-inflammatory cytokine IL-10. Additionally, treatment with *S. cornutus* extract led to activation of Nrf2 signaling, thereby maintaining intracellular redox homeostasis and counteracting NF-κB driven inflammatory response. Importantly, *S. cornutus* extract exerted a potential anti-apoptotic activity as indicated by repressing pro-apoptotic Bax activity with the simultaneous increase of anti-apoptotic Bcl-2 level leading to decreased caspase-3 activity. Furthermore, *S. cornutus* extract administration ameliorated the histopathological changes induced by cisplatin. Importantly, the higher dose of *S. cornutus* extract (500 mg/kg) elicited the most prominent effect, as compared to the lower dose (250 mg/kg). This study suggested that such extract might be considered as a novel therapeutic strategy against cisplatin-induced hepato- and nephrotoxicity.

## Figures and Tables

**Figure 1 antioxidants-11-00819-f001:**
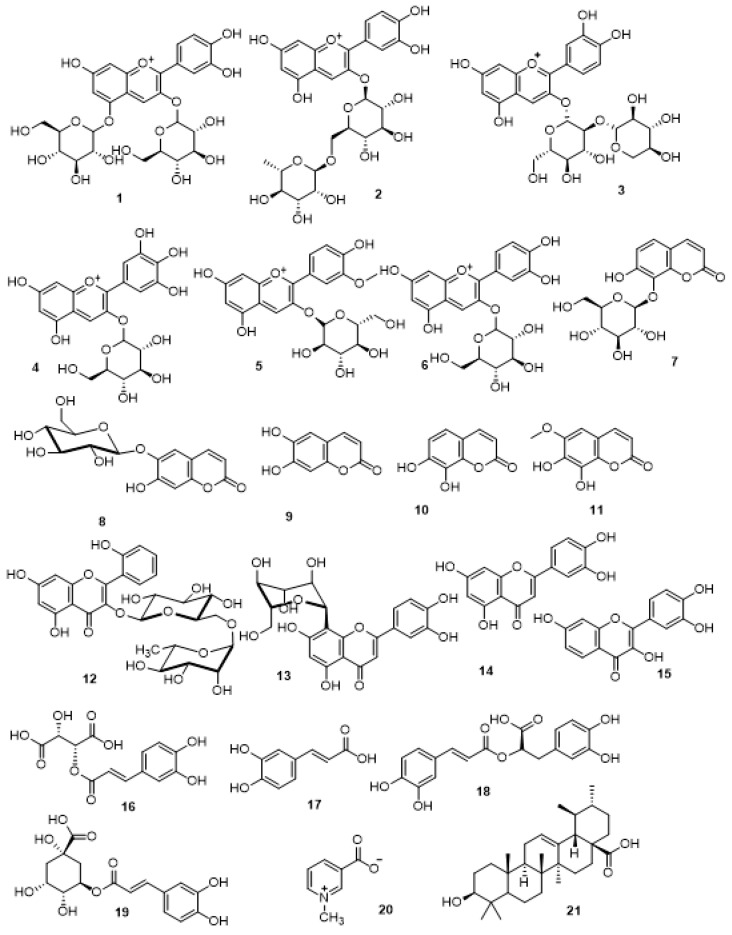
Structures of the detected metabolites as listed in Table 1.

**Figure 2 antioxidants-11-00819-f002:**
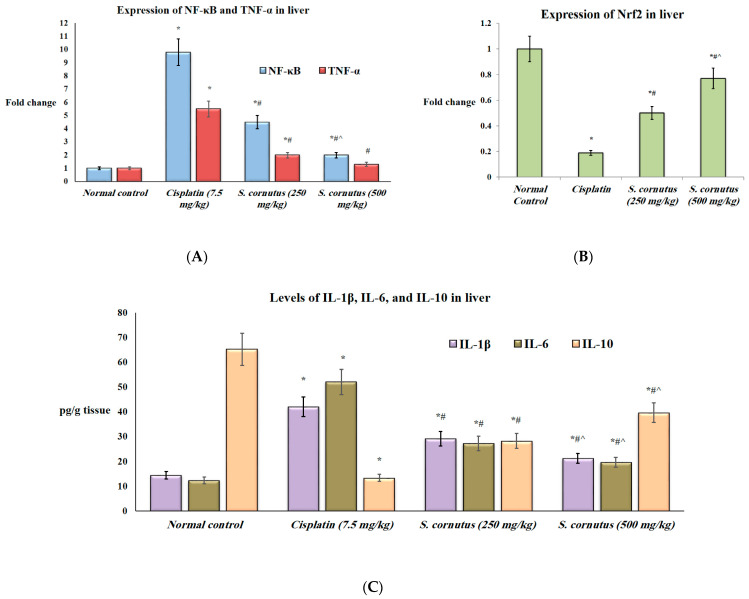
The effect of S. cornutus crude extract on the expression and protein levels of inflammtory markers in the liver tissue of the experimental mice. (**A**) NF-κβ and TNF-α expression, (**B**) Nrf2 expression, (**C**) Il-1β, IL-6, and IL-10 levels. NF-κβ = nuclear factor kappa B; TNF-α = tumor necrosis factor-α; Nrf2 = nuclear factor-erythroid factor 2-related factor 2; IL = interleukin. Data are expressed as mean ± SD and analyzed using one-way ANOVA followed by Bonferroni’s post hoc test (*n* = 8). * significantly different vs. the normal control group; ^#^ significantly different vs. the cisplatin group; ^ significantly different vs. the S. cornutus (250 mg/kg) group. Differences were considered significant at *p* < 0.05.

**Figure 3 antioxidants-11-00819-f003:**
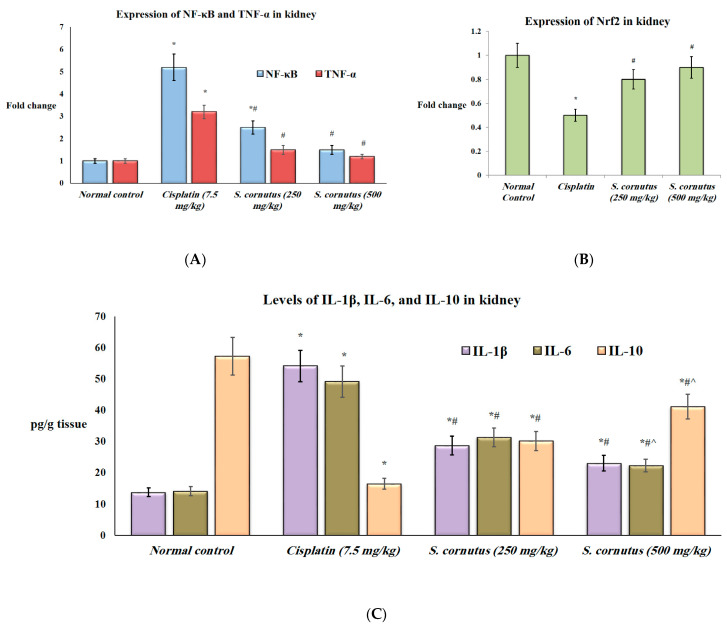
The effect of S. cornutus crude extract on the expression and protein levels of inflammtory markers in the kidney tissue of the experimental mice. (**A**) NF-κβ and TNF-α expression, (**B**) Nrf2 expression, (**C**) Il-1β, IL-6, and IL-10 levels. NF-κβ = nuclear factor kappa B; TNF-α = tumor necrosis factor-α; Nrf2 = nuclear factor-erythroid factor 2-related factor 2; IL = interleukin. Data are expressed as mean ± SD and analyzed using one-way ANOVA followed by Bonferroni’s post hoc test (*n* = 8). * significantly different vs. the normal control group; ^#^ significantly different vs. the cisplatin group; ^ significantly different vs. the S. cornutus (250 mg/kg) group. Differences were considered significant at *p* < 0.05.

**Figure 4 antioxidants-11-00819-f004:**
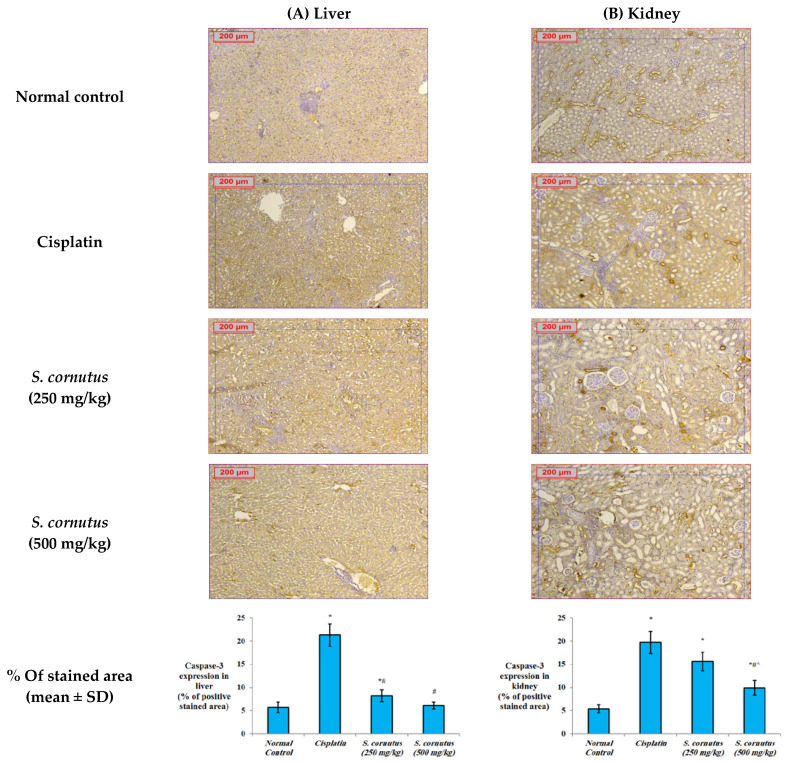
The effect of *S. cornutus* crude extract on the expression levels of caspase-3 determined by immunohistochemistry in the (**A**) liver and (**B**) kidney tissues of the experimental mice. Data of the percentage of positive stained area are expressed as mean ± SD and analyzed using one-way ANOVA followed by Bonferroni’s post hoc test (*n* = 8). * significantly different vs. the normal control group; ^#^ significantly different vs. the cisplatin group; ^ significantly different vs. the *S. cornutus* (250 mg/kg) group. Differences were considered significant at *p <* 0.05.

**Figure 5 antioxidants-11-00819-f005:**
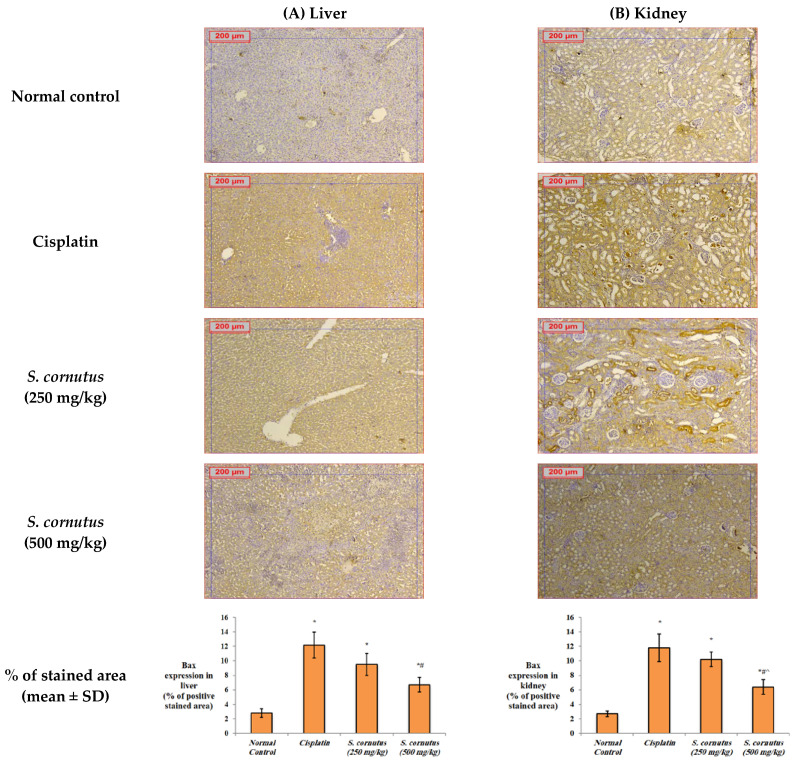
The effect of *S. cornutus* crude extract on the expression levels of Bax determined by immunohistochemistry in the (**A**) liver and (**B**) kidney tissues of the experimental mice. Data of the percentage of positive stained area are expressed as mean ± SD and analyzed using one-way ANOVA followed by Bonferroni’s post hoc test (*n* = 8). * significantly different vs. the normal control group; ^#^ significantly different vs. the cisplatin group; ^ significantly different vs. the *S. cornutus* (250 mg/kg) group. Differences were considered significant at *p <* 0.05.

**Figure 6 antioxidants-11-00819-f006:**
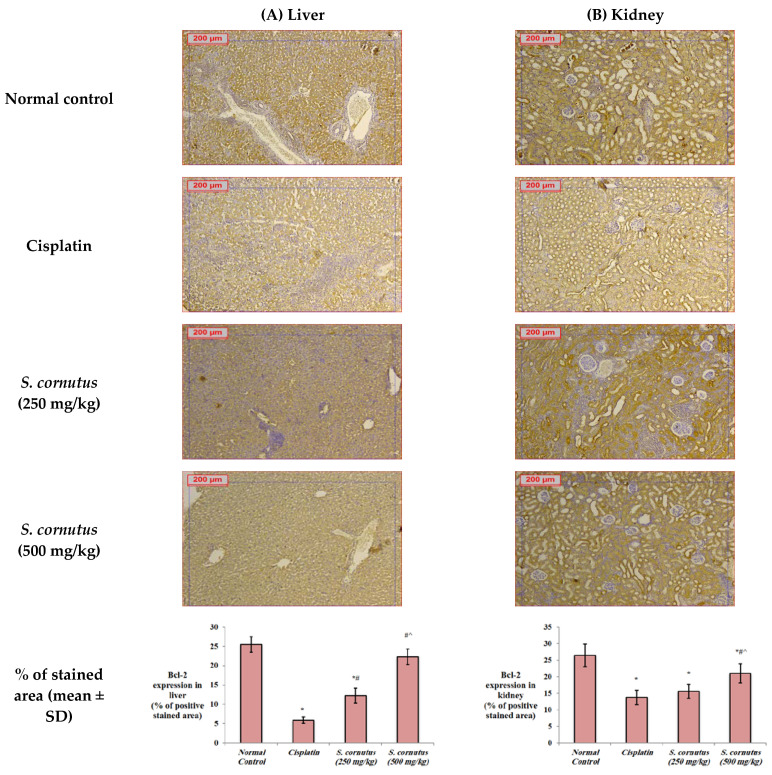
The effect of *S. cornutus* crude extract on the expression levels of Bcl-2 determined by immunohistochemistry in the (**A**) liver and (**B**) kidney tissues of the experimental mice. Data of the percentage of positive stained area are expressed as mean ± SD and analyzed using one-way ANOVA followed by Bonferroni’s post hoc test (*n* = 8). * significantly different vs. the normal control group; ^#^ significantly different vs. the cisplatin group; ^ significantly different vs. the *S. cornutus* (250 mg/kg) group. Differences were considered significant at *p <* 0.05.

**Figure 7 antioxidants-11-00819-f007:**
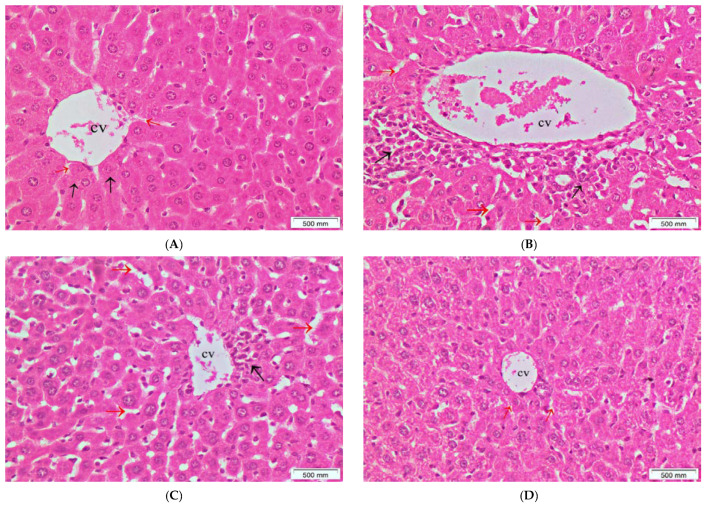
A photomicrography of liver tissue (H&E, ×200). (**A**) normal control group; shows normal hepatic architecture, with clear central vein (CV) and hepatocytes radially arranged in plate (black arrows), and normal sinusoidal spaces between them (red arrows). (**B**) cisplatin group; shows infiltration of inflammatory cells (black arrows), with congested central vein (CV) and dilated sinusoidal spaces (red arrows). (**C**) *S. cornutus* (250 mg/kg) group; shows partial improvement, less infiltration of inflammatory cells (black arrows), but still dilated sinusoidal spaces (red arrows). (**D**) *S. cornutus* (500 mg/kg) group; shows the best improvement, nearly normal hepatic architecture, and restoration of normal sinusoidal spaces (red arrows).

**Figure 8 antioxidants-11-00819-f008:**
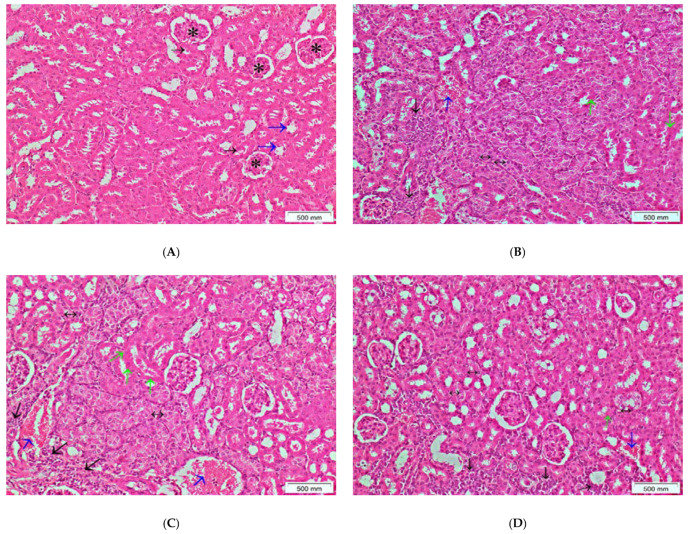
A photomicrography of kidney tissue (H&E, ×100). (**A**) normal control group; shows normal histology, with normal looking glomeruli (asterisks), proximal tubules (black arrows), and distal tubules (blue arrows). (**B**) cisplatin group; shows patchy interstitial inflammatory cell infiltrate (black arrows), congestion (blue arrows), focal tubular cell necrosis with condensed eosinophilic cytoplasm (green arrows), and regenerative changes with slightly basophilic cytoplasm and enlarged nuclei (double headed arrows). (**C**) *S. cornutus* (250 mg/kg) group; shows patchy interstitial inflammatory cell infiltrate (black arrows), congestion (blue arrows), focal tubular cell necrosis with condensed eosinophilic cytoplasm (green arrows), and regenerative changes with slightly basophilic cytoplasm and enlarged nuclei (double headed arrows). (**D**) *S. cornutus* (500 mg/kg) group; shows patchy interstitial inflammatory cell infiltrate (black arrows), mild congestion (blue arrows), minimal focal tubular cell necrosis (green arrows), and some regenerative changes with slightly basophilic cytoplasm and enlarged nuclei (double headed arrows).

**Table 1 antioxidants-11-00819-t001:** Metabolic profiling of the crude extract of *S*. *cornutus* using LC-MS/MS analysis.

No	Polarity Mode	MZmine ID	Ret. Time (min)	Measured *m/z*	Calculated *m/z*	Mass Error (ppm)	Adduct	Molecular Formula	MS/MS Spectrum	Deduced Compound	Ref.
I. Anthocyanidin glycosides											
1.	Positive	2385	5.43	611.1656	611.1607	8.02	M ^+^	C_27_H_31_O_16_ ^+^	611, 449, 287	Cyanidin-3, 5-di-O-glucopyranoside	[22]
2.	Positive	2563	6.33	595.1599	595.1657	−9.75	M ^+^	C_27_H_31_O_15_ ^+^	595, 433, 287	Cyanidin-3-O-rutinoside	[22]
3.	Positive	2660	6.49	581.1516	581.1501	2.58	M ^+^	C_26_H_29_O_15_ ^+^	581, 449, 287	Cyanidin-3-O-(2″-O-*β*-xylopyranosyl-*β*-glucopyranoside)	[22]
4.	Positive	2672	6.51	465.1022	465.1028	−1.29	M ^+^	C_21_H_21_O_12_ ^+^	465, 303	Delphinidin-3-glucoside	[23]
5.	Positive	3059	7.81	463.1282	463.1235	10.15	M ^+^	C_22_H_23_O_11_ ^+^	463, 301	Peonidin-3-O-glucoside	[24]
6.	Positive	3103	7.99	449.1082	449.1078	0.89	M ^+^	C_21_H_21_O_11_ ^+^	449, 287	Cyanidin-3-O-glucoside	[22]
II. Coumarins and their glycosides											
7.	Positive	2122	3.73	341.0866	341.0873	−2.05	[M + H] ^+^	C_15_H_16_O_9_	341, 179, 133	Daphnetin-8- glucopyranoside	[25]
8.	Negative	2357	3.79	339.0732	339.0716	4.72	[M − H] ^−^	C_15_H_16_O_9_	339, 177	Aesculin	[11]
9.	Negative	2547	5.07	177.0192	177.0188	2.26	[M − H] ^−^	C_9_H_6_O_4_	177, 89	Aesculetin	[26]
10.	Positive	3035	5.75	179.0356	179. 0344	6.70	[M + H] ^+^	C_9_H_6_O_4_	179, 133, 77	Daphnetin	[25]
11.	Negative	2757	6.01	207.0293	207.0293	0.0	[M − H] ^−^	C_10_H_8_O_5_	207, 192	Fraxetin	[26]
III. Flavonoids and their glycosides											
12.	Negative	2910	6.52	593.1481	593.1506	−4.21	[M − H] ^−^	C_27_H_29_O_15_	593, 285	Datiscin	[10]
13.	Negative	3045	6.85	447.0922	447.0927	−1.12	[M − H] ^−^	C_21_H_20_O_11_	447, 327	Orientin	[11]
14.	Negative	3564	9.43	285.0404	285.0399	1.75	[M − H] ^−^	C_15_H_10_O_6_	285, 133	Luteolin	[11]
15.	Positive	3505	9.72	287.0574	287.0556	6.27	[M + H] ^+^	C_15_H_10_O_6_	287, 269, 241, 213, 149, 137	Fisetin	[11]
IV. Phenolic derivatives											
16.	Negative	383	1.14	311.0427	311.0403	7.72	[M − H] ^−^	C_13_H_12_O_9_	311, 133	Caftaric acid	[27]
17.	Negative	973	1.24	179.0355	179.0344	6.14	[M − H] ^−^	C_9_H_8_O_4_	179, 135, 134	Caffeic acid	[11]
18.	Negative	2496	4.62	359.0758	359.0767	−2.51	[M − H] ^−^	C_18_H_16_O_8_	359, 161	Rosmarinic acid	[28]
19.	Negative	2536	4.88	353.0860	353.0873	−3.68	[M − H] ^−^	C_16_H_18_O_9_	353, 191	Chlorogenic acid	[29]
V. Other chemical classes											
20.	Positive	1680	1.39	138.0546	138.0555	−6.52	[M + H] ^+^	C_7_H_7_NO_2_	138, 94	Trigonelline	[10]
21.	Negative	4434	22.81	455.3539	455.3525	3.07	[M − H] ^−^	C_30_H_48_O_3_	455	Ursolic acid	[11]

**Table 2 antioxidants-11-00819-t002:** Total phenolics and total flavonoids content in *S. cornutus*.

Crude Extract	Total Phenolics (mg GAE/gm)	Total Flavonoids (mg QE/gm)
*S. cornutus*	206.28 ± 14.64	45.56 ± 1.78

GAE: gallic acid equivalent; QE: quercetin equivalent.

**Table 3 antioxidants-11-00819-t003:** Total antioxidant capacity, ferric reducing power activity, and DPPH free radical scavenging activity of *S. cornutus*.

Samples	Total Antioxidant Capacity (mg GAE/gm)	Ferric Reducing Power (mM Fe^+2^/gm)	IC_50_ of DPPH Scavenging Activity (µg/mL)
*S. cornutus* crude extract	49.06 ± 3.62	1.92 ± 0.71	16.10 ± 2.14
Ascorbic acid	69.32 ± 4.51	3.14 ± 0.82	10.64 ± 0.82

GAE: gallic acid equivalent; DPPH: 1,1-diphenyl-2-picrylhydrazyl.

**Table 4 antioxidants-11-00819-t004:** The effect of *S. cornutus* crude extract on the liver and kidney functions in the experimental mice.

Parameter	Normal Control	Cisplatin (7.5 mg/kg)	*S. cornutus* (250 mg/kg)	*S. cornutus* (500 mg/kg)
ALT (U/L)	51.55 ± 5.60	98.11 ± 10.10 *	76.13 ± 8.20 *^#^	63.12 ± 6.22 *^#^^
AST (U/L)	45.93 ± 5.50	92.77 ± 9.98 *	77.37 ± 8.23 *^#^	60.21 ± 7.05 *^#^^
ALP (U/L)	91.54 ± 8.73	280.38 ± 21.39 *	210.20 ± 18.28 *^#^	140.28 ± 15.30 *^#^^
Creatinine (mg/dL)	0.16 ± 0.03	0.44 ± 0.09 *	0.35 ± 0.07 *^#^	0.22 ± 0.05 ^#^^
BUN (mg/dL)	45.34 ± 3.18	79.90 ± 9.23 *	63.36 ± 7.22 *^#^	51.29 ± 5.82 ^#^^

Data are expressed as mean ± SD and analyzed using one-way ANOVA followed by Bonferroni’s post hoc test (*n* = 8). ALT = alanine aminotransferase; AST = aspartate aminotransferase; ALP = alkaline phosphatase; BUN = blood urea nitrogen; * significantly different vs. the normal control group; ^#^ significantly different vs. the cisplatin group; ^ significantly different vs. the *S. cornutus* (250 mg/kg) group. Differences were considered significant at *p <* 0.05.

**Table 5 antioxidants-11-00819-t005:** The effect of *S. cornutus* crude extract on the levels of oxidative stress markers in the liver and kidney tissues of the experimental mice.

Parameter	Organ	Normal Control	Cisplatin (7.5 mg/kg)	*S. cornutus* (250 mg/kg)	*S. cornutus* (500 mg/kg)
MDA (nmol/g tissue)	Liver	40.4 ± 5.1	287.7 ± 32.5 *	182.7 ± 21.3 *^#^	136.3 ± 15.4 *^#^^
	Kidney	122.4 ± 12.3	558.3 ± 63.3 *	304.2 ± 32.3 *^#^	162.4 ± 18.3 *^#^^
ROS (U/g tissue)	Liver	17.2 ± 2.0	56.1 ± 5.9 *	29.8 ± 3.0 *^#^	23.4 ± 2.5 *^#^^
	Kidney	15.9 ± 1.8	55.3 ± 5.8 *	33.1 ± 3.4 *^#^	25.8 ± 2.8 *^#^^
GSH (ng/mg tissue)	Liver	70.2 ± 8.1	21.3 ± 2.1 *	38.9 ± 4.2 *^#^	54.6 ± 6.3 *^#^^
	Kidney	59.2 ± 6.5	17.8 ± 1.9 *	28.7 ± 3.1 *^#^	41.2 ± 4.2 *^#^^
GSSG (ng/mg tissue)	Liver	19.7 ± 2.2	67.3 ± 7.5 *	36.5 ± 3.8 *^#^	28.7 ± 3.2 *^#^^
	Kidney	20.1 ± 2.2	63.2 ± 6.8 *	38.4 ± 4.0 *^#^	27.6 ± 2.8 *^#^^
GSH/GSSG ratio	Liver	3.6 ± 0.5	0.3 ± 0.1 *	1.1 ± 0.1 *^#^	1.9 ± 0.2 *^#^^
	Kidney	2.9 ± 0.4	0.3 ± 0.1 *	0.7 ± 0.1 *^#^	1.5 ± 0.2 *^#^^
SOD (U/g tissue)	Liver	698.9 ± 73.3	270.4 ± 30.2 *	398.2 ± 42.1 *^#^	470.7 ± 45.3 *^#^
	Kidney	187.3 ± 2.2	117.2 ± 12.3 *	125.2 ± 14.2 *	150.3 ± 16.2 *^#^
Catalase (U/g tissue)	Liver	16.8 ± 2.3	5.1 ± 0.7 *	9.2 ± 1.1 *^#^	13.2 ± 1.5^#^^
	Kidney	18.9 ± 2.5	8.1 ± 0.9 *	10.9 ± 1.2 *	15.3 ± 1.7^#^^
NO (µmol/g tissue)	Liver	11.3 ± 1.6	44.5 ± 5.4 *	23.5 ± 2.8 *^#^	18.6 ± 2.0 *^#^
	Kidney	8.6 ± 0.9	27.2 ± 3.2 *	21.3 ± 2.6 *	14.6 ± 1.5 *^#^^
iNOS (ng/g tissue)	Liver	3.5 ± 0.4	16.5 ± 2.0 *	10.4 ± 1.4 *^#^	7.6 ± 0.9 *^#^^
	Kidney	3.4 ± 0.5	11.2 ± 1.3 *	7.7 ± 0.9 *^#^	5.9 ± 0.7 *^#^

Data are expressed as mean ± SD and analyzed using one-way ANOVA followed by Bonferroni’s post hoc test (*n* = 8). MDA = malondialdehyde; ROS = reactive oxygen species; GSH = reduced glutathione; GSSG = oxidized glutathione; SOD = superoxide dismutase; NO = nitric oxide; iNOS = inducible nitric oxide synthase; * significantly different vs. the normal control group; ^#^ significantly different vs. the cisplatin group; ^ significantly different vs. the *S. cornutus* (250 mg/kg) group. Differences were considered significant at *p <* 0.05.

## Data Availability

Data is available within the article.

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
