# Peer review of "Metabolic Profiling, Chemical Composition, Antioxidant Capacity, and In Vivo Hepato- and Nephroprotective Effects of Sonchus cornutus in Mice Exposed to Cisplatin"

_antioxidants, 2022, doi:10.3390/antiox11050819_

Round 1
Reviewer 1 Report
After carefully reading the manuscript my opinion is that the information provided in this paper is significant to a certain extent and is presented in a well-structured manner. However, I have some comments, recommendations and questions:
- The English language used throughout the manuscript needs some improvements in terms of style and grammar.
- Some explanations related to the performed statistical analysis are necessary. Taking into consideration that in the manuscript is specified that only three replicate samples were analysed and in the captions of figures 2 – 8 it is mentioned that n = 8, please give some details regarding the statistical analysis and its significance / relevance.
- Too many figures are presented in the manuscript, which makes it difficult to correlate the information obtained from these figures and discussed in the text. Some of them (Figures 2A-F, 3A-F, 4A-F, 5A-F) should be presented summarized, in a meaningful manner. For example, where possible, certain figures should be grouped: 2B, 2D, 2E and 2F, etc.
- Section 3.4, lines 336-411: more discussions and explanations are needed regarding the variations of different quantities/parameters shown in Figures 2-5 and presented in the text. The same comment applies for the paragraphs between lines 424 and 439, which refers to Figures 6-8, specifying the variations that occur, but without offering possible explanations.
- In Section 3.1, starting with line 292 and Section 3.4 line 559 there are discussions related to the phenolic acids (rosmarinic and chlorogenic acids), their characteristic and properties / effects being mentioned. Why isn't there a discussion about the caffeic acid, even though it is also listed in Table 1?
- The conclusions are scarce and they do not sufficiently highlight the originality and the significant contributions of the study. Therefore, they need to be reformulated / completed.
- The references must be written in the same format, in accordance with the requirements of the journal.
In conclusion, my opinion is that the manuscript needs major revision, the content of the manuscript requiring modifications and improvements before publication could be recommended.
Author Response
Dear Editor-in-Chief of Antioxidants-1683097
The authors appreciate the constructive criticism and the valuable suggestions of the reviewers and the editorial board concerning our manuscript “Antioxidants-1683097”, under title of “Metabolic Profiling, Chemical Composition, Antioxidant Capacity and In Vivo Hepato- and Nephroprotective Effects of Sonchus cornutus in Mice Exposed to Cisplatin”
Hereby, the point-by-point reply for the raised corrections and suggestions by the reviewer #1. The authors confirm that they made all the required changes by the referees and were written in red font in the manuscript.
I hope the manuscript in its present form is eligible for publication in your journal.
Please find the attached PDF file

Reviewer 2 Report
In the article “Metabolic Profiling, Chemical Composition, Antioxidant Capacity and In Vivo Hepato- and Nephroprotective Effects of Sonchus cornutus in Mice Exposed to Cisplatin”, the authors confirmed Sonchus cornutus had antioxidants activity in vitro and in vivo. In mice, the author revealed that Sonchus cornutus could significantly ameliorate the cisplatin induced disturbances in liver and kidney functions, relieves oxidative stress, reverses apoptosis, reduces inflammation levels in liver and kidney. Furthermore, in this article, the experimental design is reasonable and the data analysis is correct, which lays the foundation for solving the liver and kidney injury caused by cisplatin. I agree to accept this manuscript after a minor revision. Here are some suggestions:
- In mouse liver and kidney, the authors demonstrate that Sonchus cornutus could reverse cisplatin-induced cell apoptosis. In cancer cell line, could Sonchus cornutus also play the same role?
- In Figure 2 and 3, the author detected the content of GSH. As we know, the content of GSSG is also an important molecule for evaluating antioxidants, and the level of GSH/GSSG could better reflect antioxidant capacity.
- The authors demonstrated that Sonchus cornutus could exert antioxidant effects in liver and kidney tissues. However, the level of ROS in blood or liver and kidney tissues were not detected.
Author Response
Dear Editor-in-Chief of Antioxidants-1683097
The authors appreciate the constructive criticism and the valuable suggestions of the reviewers and the editorial board concerning our manuscript “Antioxidants-1683097”, under title of “Metabolic Profiling, Chemical Composition, Antioxidant Capacity and In Vivo Hepato- and Nephroprotective Effects of Sonchus cornutus in Mice Exposed to Cisplatin”
Hereby, the point-by-point reply for the raised corrections and suggestions by the reviewer #2. The authors confirm that they made all the required changes by the referees and were written in red font in the manuscript.
I hope the manuscript in its present form is eligible for publication in your journal.
Please find the attached PDF file

Round 2
Reviewer 1 Report
After carefully reading the revised manuscript and the responses given by the authors to each of my previous comments, I ascertained that the authors took into consideration my suggestions, completed the manuscript and made modifications. Thus, I noticed an improvement of the manuscript, significant information being added.
In conclusion, my opinion is that the manuscript can be accepted for publication in present form.